# Recent and projected changes in rain-on-snow event characteristics across Svalbard

Hannah Vickers[1], Priscilla Mooney[1] and Oskar Landgren[2]

[1]NORCE Norwegian Research Centre, Bergen, Norway
[2]Norwegian Meteorological Institute, Oslo, Norway

*Correspondence to*: Hannah Vickers (havi@norceresearch.no)

**Abstract.**

Rain-on-snow (ROS) events in Svalbard are occurring more frequently during the winter season due to rapid and ongoing climate warming across the Arctic. ROS events have gained increasing attention in recent decades due to their cascading impacts on the physical environment, and terrestrial and marine ecosystems that are impacted by snowmelt. While the frequency of ROS events in Svalbard has been well studied and documented, other characteristics of ROS, specifically their duration, intensity and seasonal timing have received less attention. Such characteristics are equally important to quantify due to their potential consequences for the winter snowpack and snow-dependent ecosystems. This study addresses this knowledge gap using the Copernicus Arctic Regional Reanalysis (CARRA) for the present-day analysis and km-scale climate projections from the HARMONIE Climate (HCLIM) regional model obtained by downscaling the Max Planck Institute for Meteorology Earth System Model Version 1.2 at Low Resolution. The HCLIM projections cover the near future period of 2030-2070 under the high emissions scenario SSP5-8.5. For the present climate, the results show significant and positive trends in the mean duration, intensity and total precipitation of ROS events but these are confined mainly to low-lying areas of Nordaustlandet and some areas in the east of the archipelago, while no statistically significant trend was found in the southern and western areas which typically exhibit the largest values in these characteristics. On the other hand, there are significant and positive trends in ROS frequency across most parts of the archipelago except for the highest lying glaciated areas in northern Spitsbergen and Nordaustlandet. Analysis of the HCLIM future projections showed that the largest changes relative to present day conditions in all ROS characteristics are projected to occur over the mountainous and glaciated areas in the north and northeast of the archipelago, while some low lying western coastal areas are projected to experience a decrease. Moreover, while ROS has increased most in October and May in the present climate, the future climate simulations project a substantial increase in ROS events in April, which currently experiences very few, if any, ROS events. This may lead to considerable changes in snow hydrology. The frequency of ROS is projected to increase most over high elevation and glaciated areas in October, November, April and May by 2070, but with considerable reductions in low lying areas close to the western and southern coast as well as across many valleys in central Svalbard in October and May. While km-scale models, such as the

model used here, reduces some of the uncertainty in projected changes in ROS through improved representation of important physical processes, their computational costs prohibit the use of model ensembles to address uncertainty in projected changes. We partially addressed the uncertainty associated with model parameterisation and internal climate variability by analysing two coarser-resolution, 11km-scale climate projections under the moderate emissions SSP3-7.0 scenario. Our analysis indicated that there are areas of Svalbard where the change in all ROS characteristics carries greater uncertainty, as demonstrated by the opposite direction of change exhibited in the two additional global climate models. On the other hand, all three model projections showed agreement over the increase in ROS frequency over large parts of the archipelago in the 2050-2070 period. Further work should include analysing a larger ensemble of climate projections downscaled by several RCMs for Svalbard to produce a broader range of ROS scenarios, as well as carrying out a more in-depth analysis of the changes in relative contributions of local vs. remote moisture sources to changing patterns of precipitation, and analysing the hydrological impacts associated with the changes in ROS characteristics identified in this study.

## 1 Introduction

The Arctic is warming at a rate that is three to four times the global average (Rantanen et al., 2022). This enhanced rate of warming in the Arctic, known as Arctic Amplification, is stronger in the autumn and winter (Screen and Simmonds, 2010; Zhang et al., 2021) due to ocean-atmosphere feedbacks, and results in substantial changes to the wintertime climate. Interludes of winter warming bringing rain, often referred to as rain-on-snow (ROS) events, are becoming increasingly frequent across the High Arctic Archipelago of Svalbard. These events have attracted increasing research attention during the most recent decade and as such their spatiotemporal characteristics, meteorological drivers and impacts are becoming better understood and documented using a wide range of observational approaches (e.g. Bartsch et al., 2023; Vickers et al., 2022; Serreze et al., 2021 Wickström et al., 2020; Peeters et al., 2019; Forbes et al., 2016). These studies include not only their impacts on the cryosphere, but also on terrestrial and marine/coastal ecosystems as well as society. A significant consequence of ROS events is the formation of ground ice as rain percolates through the snowpack to the ground-snow interface and refreezes. This presents a significant barrier to winter forage for reindeer populations in Finnmark and Svalbard and has in some extreme cases resulted in starvation and large die-offs (e.g., Hansen et al., 2014) as well as adaptations to foraging habits (Pedersen et al., 2021). The impact of ROS events on the snowpack is largely dependent on the characteristics of a ROS event, such as the total precipitation and duration, and thus the intensity - as well as the initial properties of the snowpack itself. Snow depth and snowpack stratigraphy is an important factor which determines how rain percolates through the snowpack and consequently, if ground ice forms following a ROS event (Peeters et al., 2019) and how surface runoff and hydrology is affected (e.g., Würzer et al., 2016). Indeed, if a ROS event is intense enough, and the snowpack is thin enough, complete ablation of the snow cover may occur, removing the wintertime insulation of permafrost as well as increasing the availability of forage to reindeer. Therefore, the timing and seasonality of ROS events is also an important factor, as this dictates the initial thickness of the snowpack being impacted.

To understand which areas are most vulnerable to ROS impacts at present and in the future, reliable datasets describing the spatial and temporal variations in ROS are crucial. Recent studies of ROS climatology in Svalbard have exploited Synthetic Aperture Radar (SAR) remote sensing, due to its sensitivity to liquid water in the snowpack (Vickers et al., 2022). This dataset was compared to ROS events detected using snow models and atmospheric reanalysis, and good agreement with the SAR dataset was obtained once the models and reanalysis datasets had been calibrated against ground observations recorded at three sites across Spitsbergen (Ny Ålesund, Longyearbyen and Hornsund). However, it was shown that different temperature thresholds were required to produce the best accuracy of ROS detection with respect to the ground observations (Vickers et al., 2024). Specifically, it was found that gridded atmospheric reanalyses provided by the Copernicus Arctic Regional Reanalysis (CARRA) dataset was able to capture the frequency of ROS events very accurately when evaluated against ground observations. Until now most ROS studies of Svalbard have concentrated on documenting ROS frequency, but little attention has been paid to their duration, intensity, and timing. Earlier analyses of downscaled global climate model (GCM) simulations have highlighted a potential threefold increase in mild weather days during the winter (October-April) season by 2100, where precipitation falls on days with temperature above freezing point (Isaksen et al., 2017) while others note an increase exceeding 20% in winter rainfall projected at Longyearbyen airport (Førland et al., 2011) with greatest changes expected in the north and northeast of the archipelago. As climate warming continues to change the wintertime climate in Svalbard, it is crucial to quantify how ROS characteristics are influenced by changes in climate, as changes in ROS characteristics will also to a large degree determine the severity of their impact on snowpack stratigraphy and properties. Moreover, it is of equal importance to advance our understanding of how these characteristics are likely to change in the coming decades, such that measures can be planned that will minimise the impacts of ROS on nature and society.

Determining possible future changes to ROS climatology across Svalbard has become feasible due to recent advances in climate modelling and high-performance computing that allow climate models to run at convection-permitting (hereafter, km) scales. These scales are important for Svalbard as its climate has a large spatial variability arising from its complex topography, coastlines, fjords, glaciers and surrounding sea ice (Hanssen-Bauer et al. 2019). The benefits of such km-scale climate projections have been demonstrated already by numerous studies (Mooney et al. 2020; Køltzow et al. 2019; Prein et al. 2015). Rain-on-snow studies benefit further from these scales as climate models at these resolutions better resolve convective processes and the important separation of precipitation into rain and snow in km-scale models is physically based as opposed to the temperature-based approaches used in coarser resolution models (Mooney and Li, 2021). Specifically for Svalbard, Landgren et al. (2025) produced 2.5 km simulations using the HCLIM-AROME regional climate model (RCM) with input from the global Earth System Model MPI-ESM1-2-LR under the future scenario SSP5-8.5 (from now on HCLIM-MPI). The fine resolution of the HCLIM-MPI simulations allows a much-improved representation of the climate of the valleys and mountains on Svalbard.

The CARRA dataset now spans more than 30 years, providing an ideal opportunity to exploit the full time series to document changes in the characteristics of ROS since 1991. We have derived parameters that include their timing/seasonality, duration, total precipitation, and intensity, as well as frequency. In addition to studying the spatial variations in these parameters, we

also quantify trends in these characteristics over climate-relevant timescales (1991-2023). Lastly, we use high resolution
climate projections from one global climate model downscaled with the HCLIM-AROME RCM to firstly estimate how well
these specific characteristics are represented in the present climate, by comparing the results to those obtained with CARRA
and how they can be expected to change under a warming climate until 2070.

## 2 Methods and Datasets

### 2.1 Study area

The Svalbard archipelago is located in the North Atlantic Ocean, spanning latitudes between 74 and 81°N and comprises five
main islands, with Spitsbergen being the largest island (Fig.1). Wintertime sea ice is found just north and north-east of
Svalbard, while to the west of the archipelago is the West Spitsbergen Current. The climate of Svalbard is therefore heavily
affected by the location of the sea ice edge, contributing to a strong a southwest-northeast gradient, with milder coastal climates
in the west and south, and cold inland climate influenced by sea ice presence and variability in the north and east (Day et al.,
2012). Annual precipitation at Longyearbyen airport, in the central part of Svalbard is low, and ranges from 121.8mm (2021)
to 310mm (2016), while Ny Ålesund in the northwest part of Svalbard, experiences a substantially wetter climate with annual
precipitation ranging from 205mm (2019) to 749mm (2018). The occurrence of wintertime ROS events is reflected by the
climatic gradient, with highest frequency in the south and west and very few events per winter in the north and east (e.g.,
Wickström et al., 2020; Vickers et al., 2022, 2024).

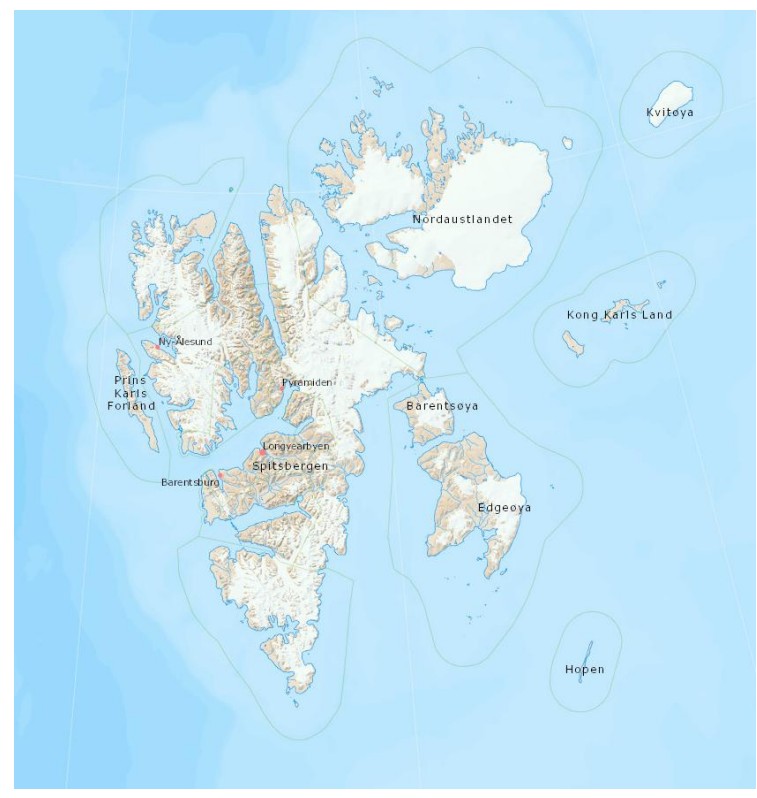


**Figure 1: Overview of the Svalbard archipelago, showing the main islands (Spitsbergen, Nordaustlandet, Edgeøya, Barentsøya). Bjørnøya lies farthest south and is not shown (source: https://toposvalbard.npolar.no/ courtesy of the Norwegian Polar Institute).**

**2.2 CARRA dataset**

In this study we use data from the East domain of the Copernicus Arctic Regional Reanalysis (CARRA) dataset, which covers all of Svalbard and its surrounding waters. CARRA provides 3-hourly reanalyses and short-term hourly forecasts of atmospheric and surface meteorological variables at 2.5 km resolution (Schyberg et al., 2020). Earlier evaluations of CARRA have already demonstrated its added value compared to other reanalysis datasets for Svalbard (Køltzow et al., 2022). Following the approach outlined by Vickers et al. (2024), we use the 2m air temperature and snow water equivalent (SWE) reanalyses at 3-hourly resolution and averaged the data to daily values. Forecasted precipitation data at lead times of +6 and +30 hours with initial time 00UT were obtained and the difference was used to calculate the 24-hour accumulated precipitation values from 0600 UTC to 0600 UTC the following day. CARRA data were obtained from 1991 to 2023 to analyse trends for the present-day climate, while only a part of the dataset overlapping with the historical period of the HCLIM simulations was used for model evaluation (2000-2020). Based on the calibration approach described in the earlier study by Vickers et al., 2024, a rain-on-snow *day* was detected when the daily mean 2m temperature was >-0.5°C, daily precipitation was >1mm and SWE was >2mm, since these thresholds produced the highest accuracy of ROS day detection when evaluated against in-situ observations.

## 2.3 HCLIM climate model data

This dataset consists of km-scale climate simulations of Svalbard covering the period 1991-2070 and is detailed in Landgren et al. (2025). The data was produced by dynamically downscaling the Max Planck Institute for Meteorology Earth System Model Version 1.2 at Low Resolution (MPI-ESM1-2-LR, Gutjahr et al., 2019) under SSP5-8.5 with the HARMONIE Climate (HCLIM, Belušić et al. 2020, Wang 2024) cycle 43 regional climate model to 2.5 km horizontal grid spacing using nonhydrostatic convection-permitting HARMONIE-AROME atmospheric physics.

For evaluation of present climate conditions, the dataset also consists of a dynamical downscaling of the 5th generation ECMWF Reanalysis (ERA5, Hersbach et al. 2020) with HCLIM for the same domain and resolution but only covering the period 2000-2020. The HCLIM configuration includes the SURFEX land-surface model with ISBA Explicit Snow scheme. More details and evaluation of the 2.5 km simulations and their evaluation over Svalbard are available in Landgren et al. (2025). From here on the HCLIM simulations produced by downscaling the ERA5 Reanalysis data will be referred to as HCLIM-ERA5 and the data produced by downscaling MPI-ESM1-2-LR will be referred to as HCLIM-MPI for clarity.

To identify ROS days in the HCLIM-ERA5 simulations for comparison with the CARRA results for the present climate (2000-2020), we applied the same thresholds as was used for the CARRA dataset, to the variables $pr$, $tas$ and $snd$, where $pr$ is the accumulated precipitation, which includes both solid and liquid precipitation, $tas$ is the 2m air temperature and $snd$ is the snow depth water equivalent. All variables are available at 3-hourly intervals, but for the purpose of producing comparable results to CARRA, we have produced daily mean values for the 2m air temperature and snow depth water equivalent variables, and a total daily precipitation estimate by taking the difference between the maximum values of the accumulated precipitation for the following day and the current day. In addition, a $prrain$ (accumulated rain) variable was also made available from the RCM simulation to assess the impacts of different approaches for separating rain and snow on the results. The $prrain$ variable is derived from the microphysical scheme of the RCM and uses a physics-based approach to separate snow and rain. This contrasts with the approach used in this study which separates rain from snow in the precipitation variables using a temperature threshold-based approach. To detect ROS using $prrain$, we applied the same daily precipitation threshold as we used for the CARRA total daily precipitation (1 mm) and to the snow depth water equivalent (2 mm). We have included the analysis of the ROS characteristics in the present and future climate using the $prrain$ variable in the Appendix (Fig.A2 to A5).

## 2.4 Definition of ROS event characteristics

For the purposes of this study, a rain-on-snow event is defined as consecutive rain-on-snow days where the criteria for detection were met. By using this definition, a rain-on-snow *event* can be characterised by its duration, which in turn determines the total precipitation as rain that fell during the event and thereby the average intensity of the event, given by the total precipitation divided by the duration. ROS events are detected in the period 1 October to 31 May *for each winter season*, which includes the shoulder seasons with onset and disappearance of snow cover and the mean values of ROS duration, total precipitation, and intensity are calculated for the time series. The number of ROS events are also recorded for each month of the winter

season to identify spatial variations and trends in their seasonality. Linear regression is performed on the time series to obtain annual trends using the slope of the regression line, and multiplied by ten to obtain the decadal trends, which are presented in section 3. We used the p-value returned by the scipy.stats.linregress function to determine if the trend was statistically significant or not ($p<0.05$).

## 3 Results

### 3.1 Present day climatology and trends in ROS

In Fig. 2 the trend in each ROS characteristic is shown for the full CARRA period (1991-2023) for (a) frequency, (b) duration, (c) total precipitation, and (d) intensity, while the climatological average of each characteristic is shown in Figure 2(e) to (h) for the overlapping period of the CARRA and HCLIM-ERA5 dataset (2000-2020) while the in. For all ROS variables there is a clear southwest-northeast gradient, with typically highest values of ROS frequency, duration, intensity and total precipitation found in the southern and western parts of the archipelago, while the lowest values are found in the more glaciated northern and northeastern areas. While the mean values of each ROS characteristic displayed in Fig.2 (e) to (h) show the overall climatology across the archipelago, there is a large degree of interannual variability as well as geographic variability. This can be seen in the time series of each ROS characteristic shown in Figure A1 for four arbitrary sites chosen for their contrasting locations; Hornsundneset, located at the coast in southwest Spitsbergen; Reindalen, in Nordenskiöld Land; Engelsbukta, just south of Ny Ålesund on the Brøgger peninsula in northwest Spitsbergen, and Åsgårdfonna, a high elevation glaciated region in the north of Spitsbergen.Addressing first the significant trends since 1991 (grid cells where $p < 0.05$ only), it is evident that significant and positive trends are found predominantly across eastern and north-eastern areas of the archipelago in all characteristics. For the ROS frequency (Fig. 2(a)), significant and positive trends are much more widespread than for the other ROS characteristics, and are also found in southern, western and central parts of the archipelago, as well as in the coastal parts of the northeast and across Edgeøya. Across Nordaustlandet, increases of up to 1 event per winter per decade are found around the coastal areas which include both land and glaciated parts. On Spitsbergen, positive trends of up to 2 to 2.5 events per winter per decade are occurring across Nordenskiöld Land as well as the southern parts of Spitsbergen and some areas in the northwest. ROS frequency is also increasing over most of Edgeøya. Examining the trends in the duration, total precipitation, and mean intensity of ROS events, there are also positive and significant trends, but the geographic variations are somewhat different compared to the trends in ROS frequency. For ROS duration (Fig. 2(b)), significant and positive trends are mainly confined to low-lying valley areas across eastern parts of Nordenskiöld Land and northern Spitsbergen but on Nordaustlandet positive trends are exhibited around the entire coast of the island, and in general the decadal trends are also greatest here, with increases of up to 0.5 days per event. Related to the positive trend in ROS duration is a positive trend in the total precipitation (Fig. 2(c)) and intensity (Fig. 2(d)), which is exhibited across the same areas on Nordaustlandet and eastern parts of Spitsbergen. Typical trends in these areas are of the order 2 to 4 mm per event per decade for total precipitation and around 1

to 2.5 mm/day per decade in ROS intensity. Somewhat larger increases in total precipitation of 4-5mm per event (per decade)
are found on Edgeøya and Barentsøya.
Figure 2(i) to (l) shows the climatological means of the ROS frequency, duration, total precipitation and intensity obtained
using the HCLIM-ERA5 downscaled to 2.5km. The differences between these means and the CARRA mean for the
corresponding variable are shown in panels (m) to (p). The differences are calculated as HCLIM minus CARRA. Comparing
the climatological averages of the characteristics obtained using CARRA (Fig. 2 (e) to (h)) and the 2.5km HCLIM-ERA5 (Fig.
2(i) to (l)) for the present climate, the geographical variations are reproduced reasonably well by the HCLIM-ERA5 dataset
even though the absolute values for all characteristics are lower with respect to the CARRA output for the ROS frequency and
duration. The difference is typically of the order of 1 to 2 events per winter and 0.5 to 0.75 days per event respectively, with
CARRA showing higher values over most of the archipelago, except for across the larger glaciated areas over Nordaustlandet
where HCLIM-ERA5 dataset tends to show slightly more ROS events than the CARRA simulations.

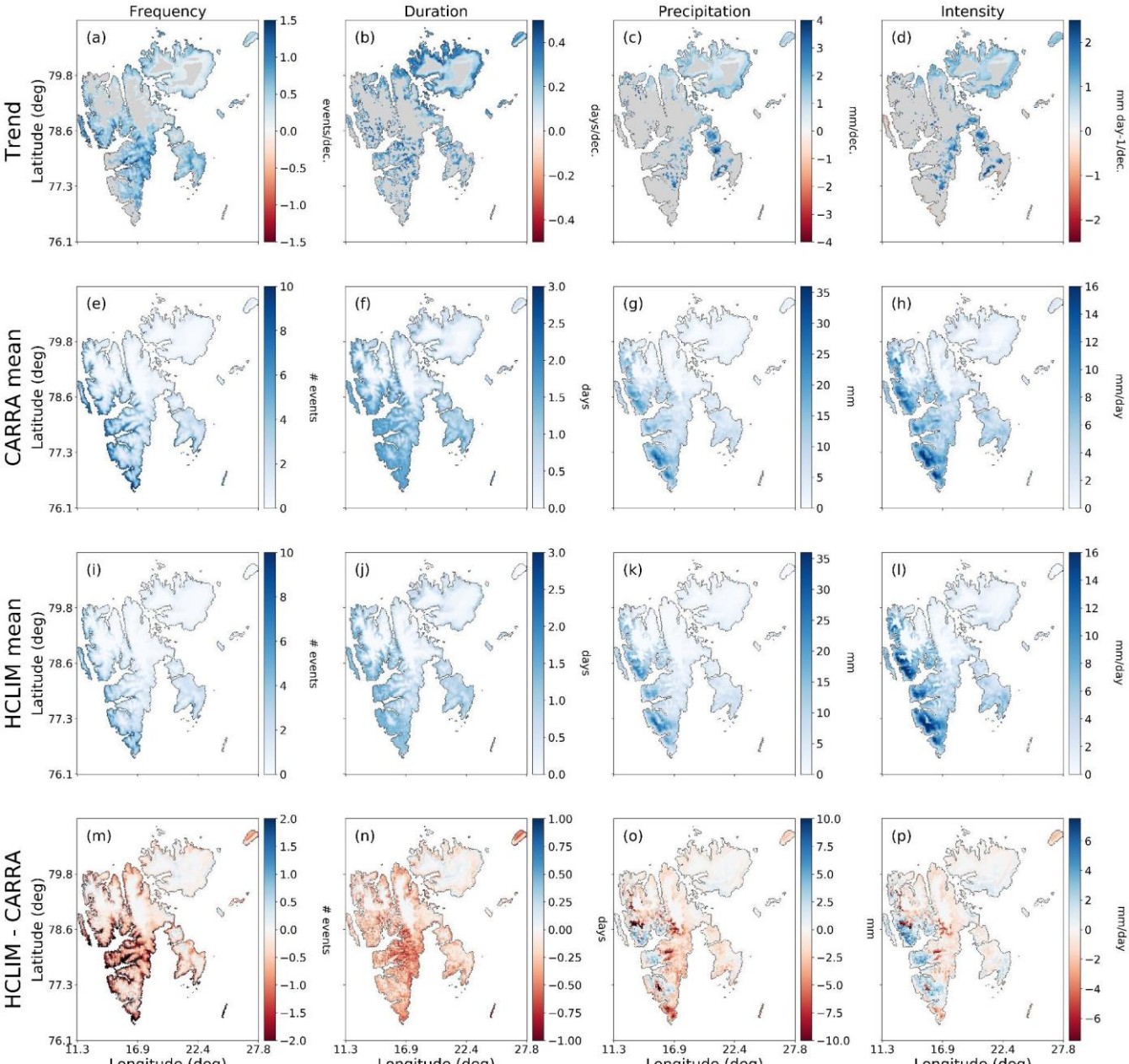

**Figure 2: Present day trends derived from the CARRA dataset (a to d) and climatology of ROS frequency, duration, total precipitation per event, and mean intensity for 2000-2020 (e) to (h). Trends are obtained for the entire CARRA period (1991-2023). ROS climatology for 2000-2020 obtained with HCLIM-ERA5 at 2.5km resolution (i to l) and the differences between HCLIM-ERA5 and CARRA (m to p).**

For ROS frequency, both CARRA and HCLIM-ERA5 show that there is a southwest-northeast gradient, with ROS occurring most frequently in the southwest and western coastal areas of Spitsbergen and decreasing across inland areas. However, while ROS are at present occurring less frequently across eastern and inland areas of the archipelago, one may note that it is these

areas where ROS have been increasing in frequency during the past 30 years (Fig. 2 (a)). The same climatic gradients are
exhibited by the averages of ROS duration, total precipitation, and intensity. However, the contrast between western and
southwestern coastal areas and inland areas of Spitsbergen is not so prominent for ROS duration as it is for total precipitation
and intensity. The mean duration of ROS events is of the order of 1-2 days across the southern, central, and north-western parts
of Spitsbergen, with an overall agreement in CARRA and HCLIM-ERA5. However, comparing the mean ROS total
precipitation and intensity, HCLIM-ERA5 tends to estimate higher total precipitation along the western coast compared to
CARRA, even though the mean event duration in these areas is only marginally lower, leading to an overall higher event
intensity in the HCLIM-ERA5 dataset in these western regions. The lower intensity in these regions in the CARRA dataset is
likely the result of slightly longer ROS durations estimated by CARRA, while total precipitation is also overall lower than
HCLIM-ERA5 in these areas. On the other hand, CARRA tends to estimate higher total precipitation across the more eastern,
southern and inland parts of the archipelago, as well as in parts of the north. However, the greatest differences between the
datasets for the mean total precipitation are typically only of the order 5 to 7 mm per ROS event.
Figure 3 decomposes the trend in ROS frequency by month using the CARRA dataset from 1991-2023. Only significant trends
($p < 0.05$) are shown, while non-significant trends are indicated by the grey shading. It is striking to note that ROS frequency
has increased significantly in predominantly three months of the winter season; mid to late autumn (October - November) and
spring (May). Less widespread and weaker trends are also observed in February. Moreover, the geographical distribution of
the significant trends is different and contrasting for these months; in October, positive trends in ROS frequency are
predominantly found around the coastal areas of Nordaustlandet, eastern and southern Spitsbergen and Edgeøya, areas
typically associated with a colder inland climate, as well as some higher elevation parts of the northwest while in February the
significant and positive trends are found only along the low lying parts of the western coast of Spitsbergen and some parts of
southern Spitsbergen, typically associated with a milder maritime climate. In May, significant and positive trends are found
over most of Nordenskiold Land, northwest and southern Spitsbergen as well as parts of Edgeøya and Barentsøya. The trends
exhibited in ROS frequency exhibited in Fig. 3 thus originate mainly from changes in ROS frequency during October,
November and May, while ROS frequency during all other months of the winter have not changed significantly since 1991.

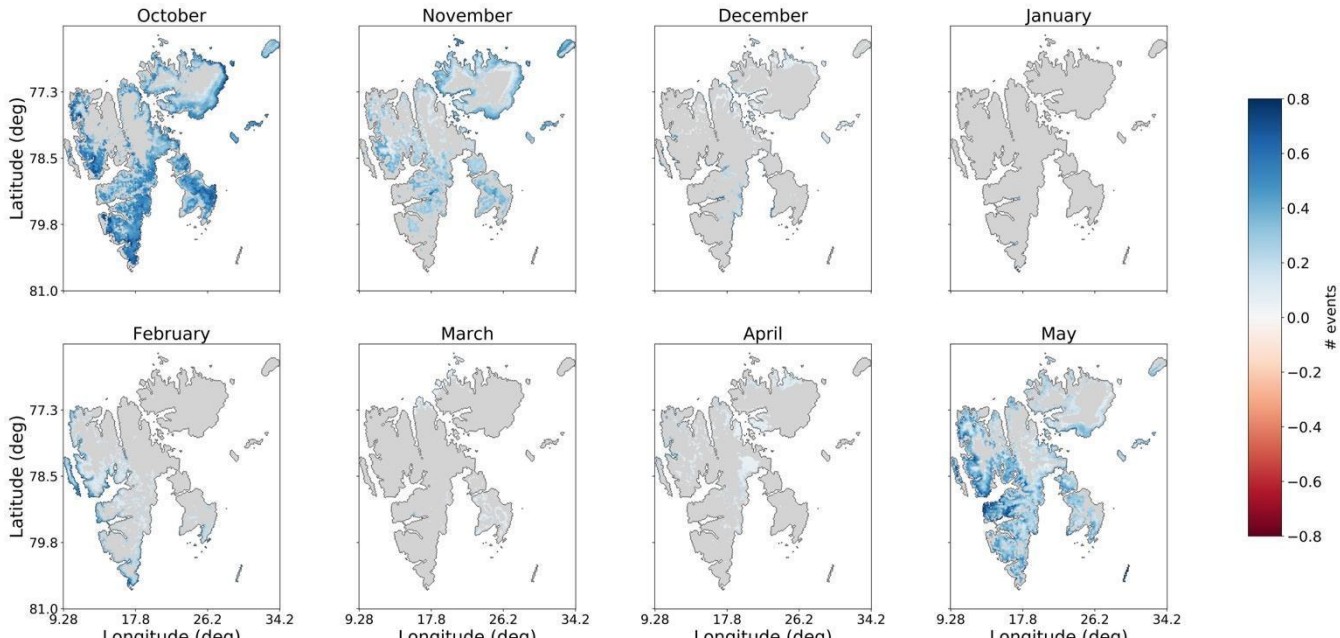


**Figure 3: Decadal trend in number of events per month from October to May for the 1991-2023 CARRA period. Only significant
trends (p < 0.05) are shown while grey areas correspond to areas with non-significant trends.**

In Sect. 3.2 we examined the projected changes in ROS during the next four decades under a high emissions scenario, obtained
using the HCLIM-downscaled climate projections from the MPI global climate model.

## 3.2 Projected future changes in ROS characteristics

Figure 4 shows the projected change in the mean number of ROS events, the mean duration, total precipitation, and intensity
in the 2030-2050 and 2050-2070 periods from the HCLIM-MPI dataset, compared to the mean for the historical period (2000-
2020). In terms of the absolute changes, Fig.4 shows that for both the future periods examined, 2030-2050 and 2050-2070,
HCLIM-MPI projects an increase in frequency of ROS events that is greatest over the areas where significant and positive
trends in frequency were identified in Fig.2. That is to say, the areas that are projected to have the greatest increase in number
of ROS events are across the eastern and southern parts of Spitsbergen, as well as across Edgeøya and Barentsøya. Northern
parts of Spitsbergen and Nordaustlandet are also projected to experience increases in ROS frequency, but to a lesser degree.
At the same time, the lowest lying coastal areas in western Spitsbergen are projected to experience a decrease in ROS frequency
in the two future periods. The change in mean number of events for the 2050-2070 period could be twice as great as for the
2030-2050 period, as indicated by the different colour scales used in Fig.4. The geographical variation in change in ROS events
is somewhat different to the geographical variations in ROS frequency for the present climate, where ROS are currently
dominant in the western part of Spitsbergen, but exhibits similarities to the geographical variations in significant trends since
1991 (Fig. 2). Figure 4 shows that the mean duration of ROS events is projected to increase by up to 1 day in the future
scenario, with the largest increases in the northern and north-eastern parts of the archipelago, as well as weaker increases in
mean ROS duration across eastern parts of Nordenskiöld Land and Edgeøya. Thus, ROS duration is projected to increase most
over glaciated and mountainous areas in the northern part of the archipelago, where there are very few ROS events in the
present climate (Fig. 2). This could be due to an overall increase in wintertime temperatures, leading to a greater probability
of days above freezing with precipitation than in the present climate. However, these areas with the greatest increase in mean
ROS event duration currently have a mean duration between 0 to 0.5 days in the present climate, indicating that there are at
present only isolated ROS events that do not occur every winter. The mean projected increase in ROS event duration is typically
only of the order of 1 day (at most) by the 2050-2070 period over these areas, while the mean increase in number of events in
the areas that exhibit greatest increase in duration is of the order of 2-3 events for the same period, indicating that the ROS
events that do occur may be single-day events. At the same time there are projected decreases in the mean duration, total
precipitation and mean intensity of events across the western coastal parts of Spitsbergen in both future periods examined,
even though there are projected to be increases in ROS frequency in these areas. For the changes in ROS total precipitation
and intensity, several specific regions stand out as having the largest projected increases; the east coast of Spitsbergen (up to
15 mm per event) and across the southern and eastern coastal areas of Nordaustlandet. Notably, the change in total precipitation
and intensity by 2050-2070 (with respect to 2000-2020) is only slightly greater than the changes in 2030-2050. This differs
from the ROS frequency and duration, where the increases are projected to be twice as great and over large areas in the 2050-
2070 period compared to 2030-2050.

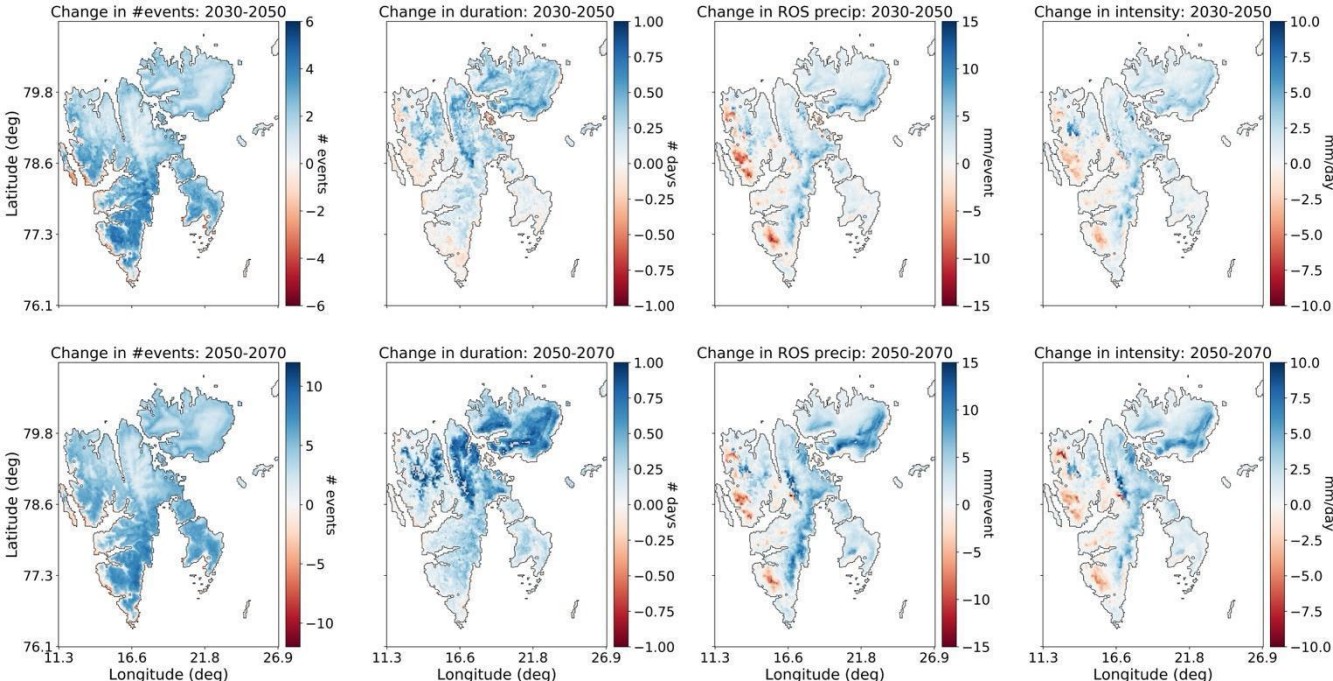

**Figure 4: Change in ROS frequency, duration, total precipitation, and mean intensity using the HCLIM-MPI dataset for 2030-2050**
**(upper row) and 2050-2070 (lower row) relative to the 2000-2020 averages.**

Figure 5 also shows the projected changes in the four ROS characteristics but expressed as the percentage change relative to
the 2000-2020 mean as opposed to the absolute values. This approach illustrates the areas that could experience the most
dramatic changes in the ROS characteristics with respect to the present-day climatology. The glaciated and mountainous
regions in northern Spitsbergen and especially across Nordaustlandet are projected to undergo the greatest change relative to
the 2000-2020 average. While the percentage changes in ROS frequency are very large for areas projected to experience the
greatest change (~300% in 2030-2050, ~700-800% in 2050-2070), it should be recalled that the mean frequency, duration,
total precipitation and intensity of ROS events across these areas are at present very low; thus, even a change producing on
average one single-day ROS event per winter results in change of several hundred percent relative to the present-day values.
Nevertheless, such considerable relative changes over glaciated areas may have considerable impacts on hydrology, which
could subsequently have consequences for fjord and marine ecosystems due to increased freshwater runoff during the
wintertime. This will be further discussed in Sect. 4.

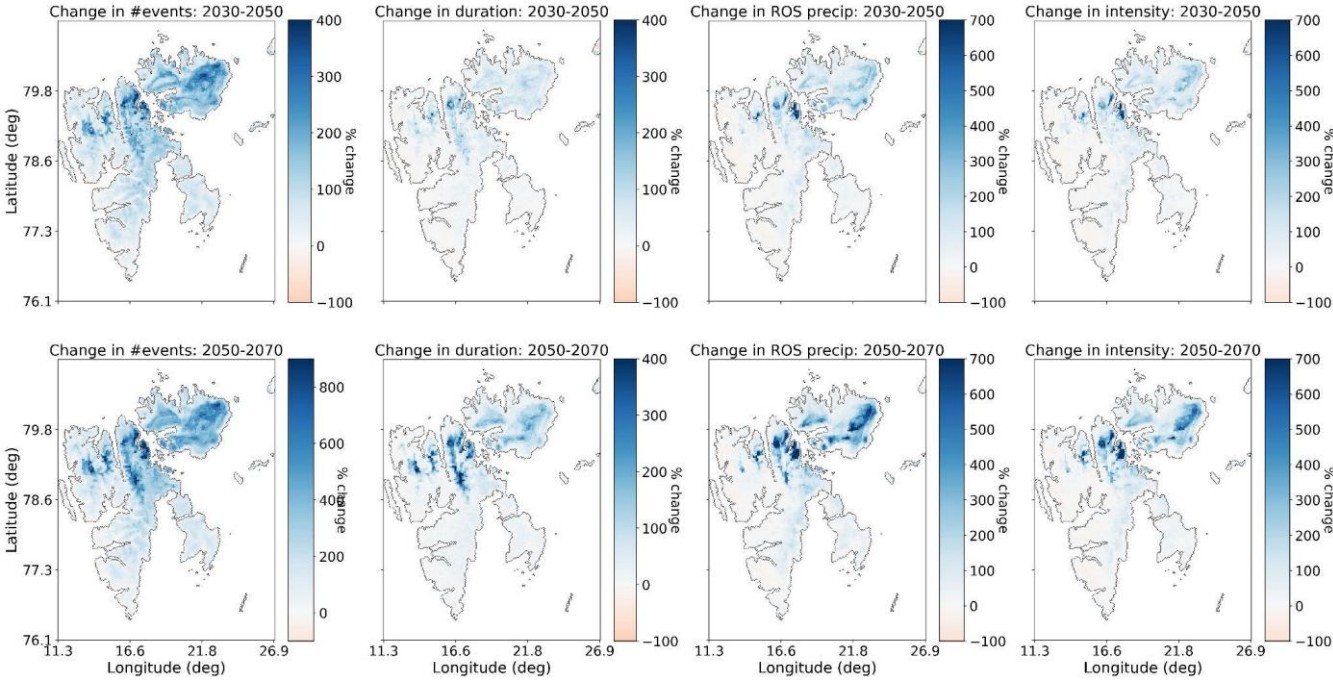


**Figure 5: Change in ROS frequency, duration, total precipitation, and mean intensity using the HCLIM-MPI dataset for 2030-2050**
**(upper row) and 2050-2070 (lower row) expressed as a percentage of the 2000-2020 averages.**
In Fig. 6 the projected changes of ROS frequency are shown for each month of the winter period (October-May). For the earlier
2030-2050 period, the greatest increase in the number of ROS events is projected to occur in October across eastern parts of
Spitsbergen, as well as across eastern parts of Nordaustlandet and on Edgeøya. Noticeable decreases of up to 2 events per
winter are projected along the entire western coast of Spitsbergen in October. A similar pattern of change is also projected for
November in the 2030-2050. Meanwhile, increases in ROS frequency during May are projected in the western and southern

parts of Spitsbergen. Smaller increases (0.5-1 events per winter) in the mean ROS frequency are projected to take place in January and February, but confined to the western, central, and southern parts of Spitsbergen. By 2050-2070 the situation changes quite dramatically. While both increases and decreases in the mean ROS frequency may continue to occur in October, November and May across the same areas as during the 2030-2050 period, increases in ROS frequency (>1.5 events per winter) are projected to occur in April, and these changes are present across large parts of the entire archipelago. Only some glaciated parts of northern Spitsbergen and Nordaustlandet are not projected to have such great increases in ROS frequency during April in the 2050-2070 period. Furthermore, changes in ROS frequency of comparable magnitude could also take place during January, a month that has until now not experienced significant changes in ROS frequency (Fig. 2). In Fig. 6 it can also be seen that there are projected to be weak decreases in the mean number of ROS events in some areas of the northern and eastern Spitsbergen in December, and across western and southern areas during March for the 2030-2050 period. However, in the later future period (2050-2070) these same areas exhibit increases with respect to 2000-2020.

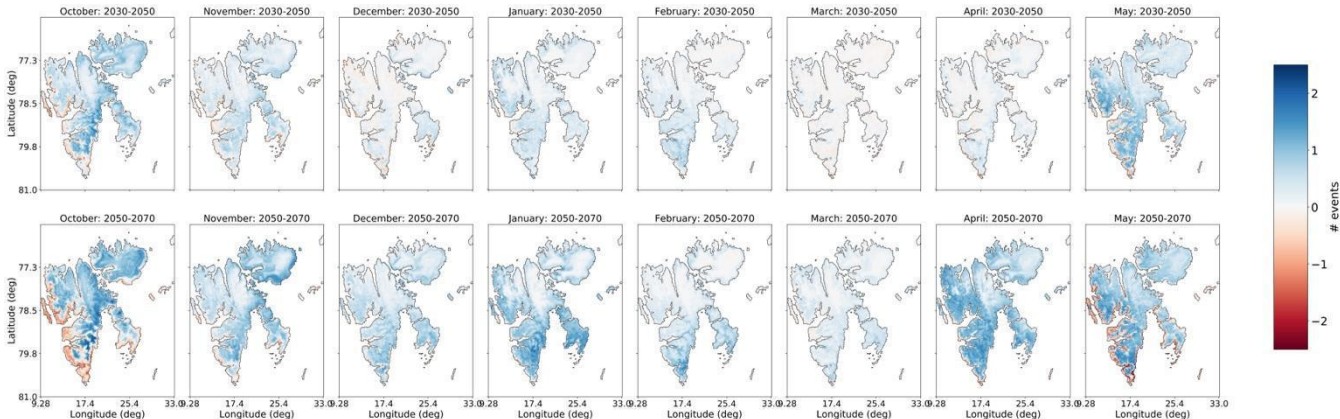

**Figure 6: Changes in ROS frequency by month from October to May using the HCLIM-MPI dataset, for 2030-2050 (upper row) and 2050-2070 relative to the reference period 2000-2020.**

Figure 7 shows the spatial distribution of the absolute change in ROS frequency, duration, total precipitation and intensity for land and glaciers. For the ROS frequency in the near-future period (2030-2050), both land and glacier areas exhibit a change ranging from -2 to +4 events per winter on average compared to 2000-2020, while for the latter period (2050-2070) this change is clearly almost doubled with a range from -4 to +10 events. For glaciers there are no areas experiencing a reduction in frequency for the two periods, while there are some areas with a decrease in frequency across land areas. There is also a clear positive shift in the distribution of change in ROS duration for glaciers compared to the change in mean ROS duration for land areas, which is most prominent for the 2053-2050 period. While the peaks of the mean duration distributions lie at roughly +0.05 and +0.3 days for land and glaciers respectively in 2030-2050, the peaks lie at +0.2 days for both land and glaciers in the 2050-2070 period, but with the distribution for glaciers skewed such that the majority of the glaciated areas exhibit increased ROS duration of up to +1.2 days in the later 2050-2070 period. Likewise, there is a shift in the peak of the distribution of the projected ROS total precipitation over glaciers with an increase of +2mm in 2030-2050 relative to the 2000-2020 period and +4mm increase by 2050-2070. For land areas, the overall change in the mean total precipitation has a peak at +1mm and

+2mm for each future period respectively. For ROS intensity, the mean change lies at approximately +2mm/day in both future
periods and the distribution of changes for land and glaciers overlap. While the change in ROS frequency over glaciers is
mostly positive, it can be seen that there are both positive and negative changes in the ROS duration, intensity and total
precipitation for both land and glaciers.

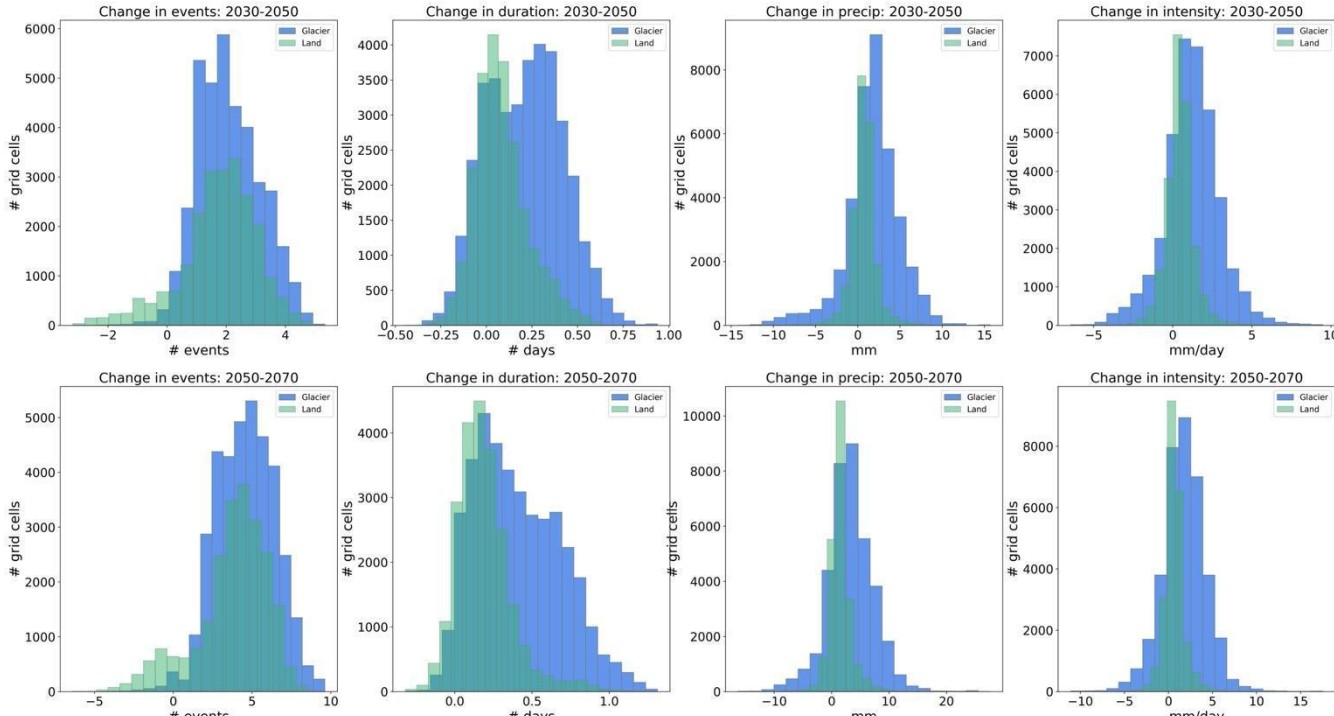

**Figure 7: Spatial distribution of change in ROS frequency, duration, total precipitation and mean intensity for 2030-2050 (upper**
**row) and 2050-2070 (lower row) shown for land and glaciers separately.**

**4 Discussion**

This study has utilized a 32-year time series of an atmospheric reanalysis and a 70-year time series of a km-scale climate
simulation for Svalbard to (i) study how four characteristics of ROS events have changed in the most recent three decades
from 1991 to 2022, and (ii) how they may change in the forthcoming decades, specifically, 2030-2050 and 2050-2070, as
represented by one km-scale regional climate model simulation under a high greenhouse gas emissions scenario.

## 4.1 Analysis of ROS characteristics in the present climate

Statistically significant and positive trends in all parameters were found predominantly in the lower-lying outskirts of Nordaustlandet in the north-east of the archipelago, and across Edgeøya in the southeast, while significant increases in frequency were also found across the eastern areas of Spitsbergen. No significant trends in mean ROS duration, intensity and total precipitation were found in the western parts of Nordenskiöld Land and southwest Spitsbergen, where ROS frequency is at present greatest. That is to say, the largest and significant increases in ROS frequency, duration, total precipitation and intensity are found in areas where there are at present very few, if any, ROS during the winter months.

Positive trends were exhibited by all ROS characteristics since 1991 using the CARRA dataset, but significant trends are confined to Nordaustlandet and Edgeøya, areas that are in closest proximity to sea ice presence during the winter. Sea ice extent around Svalbard has been declining (e.g., Onarheim et al., 2014) and it has been shown to be a key driver of strong warming trends across Spitsbergen (Isaksen et al., 2016, Isaksen et al., 2022) in recent decades. Wickström et al. (2020) found that local and regional sea ice extent drives seasonal variations in temperature and precipitation patterns across Svalbard, but the authors found less sensitivity of ROS to sea ice presence, as they found that ROS are primarily driven by the advection of warmer air masses from the southerly sector. Other studies of ROS events across the Yamal and Alaska have, on the other hand, noted below average sea ice concentration prior to events, suggesting a link between sea ice concentration and ROS (Forbes et al., 2016; Bartsch et al., 2023). While it has not been the objective of this study to quantify what role sea ice presence plays in the occurrence and observed trends in characteristics of ROS events, this could be an important topic to consider for future studies, given the rapid and ongoing decline of sea ice around Svalbard and across the polar regions, and its potential impact on atmospheric circulation and precipitation patterns.

Comparing the ROS characteristics of the CARRA dataset to those produced for the same period using the regional climate model driven with ERA5, it was found that the regional climate model projections provided by the HCLIM-ERA5 dataset reproduces the overall geographical variations in ROS frequency, duration, total precipitation, and intensity for the present/historical climate qualitatively well. However, the HCLIM-ERA5 datasets estimates slightly lower ROS frequency and duration compared to CARRA dataset over most of the archipelago. More conflicting results were found for the total precipitation and intensity, where HCLIM-ERA5 tended to estimate somewhat higher total precipitation across the western coast and southern parts of Nordaustlandet, but lower total precipitation across the rest of the archipelago. A comprehensive evaluation of the 2.5km-scale HCLIM-ERA5 simulations has recently been carried out (Landgren et al., 2025) and indicates that compared to CARRA, HCLIM tends to produce a slightly colder bias over land over most of Spitsbergen, but slightly warmer bias over eastern areas due to lack of snow cover over sea ice in ERA5 in the Barents sea region. This certainly could contribute to the differences we observed in the ROS characteristics between CARRA and HCLIM-ERA5, with underestimations in the characteristics by HCLIM over Spitsbergen but smaller differences over Nordaustlandet. Overall, the comparison of ROS climatology for the historical period demonstrates that the future projections of ROS characteristics should produce reliable climatologies. An additional source for the discrepancy in absolute values of the characteristics for the present

climate (2000-2020) could be due to the uncertainty in temperature threshold for partitioning rain and snow used in the different datasets. The temperature threshold for the CARRA dataset was determined via calibration against ground observations, and different datasets have been shown to require different temperature thresholds for detecting ROS (Vickers et al., 2024) whereas a calibration has not been done for the HCLIM-ERA5 simulations; therefore, applying the CARRA threshold to the HCLIM-ERA5 dataset cannot be expected to reproduce identical results.

Our analyses based on CARRA indicate that there have been significant and positive trends in ROS frequency since 1991 during October, November, February and May, and that the geographical variations in these trends are not the same in these months. While for the late autumn/early winter ROS the greatest trends have occurred over Nordaustlandet and eastern areas of Spitsbergen, in mid-winter (February) the trends in ROS frequency have been strongest across western and southern parts of Spitsbergen.

## 4.2. Future projections of ROS characteristics

In the two future periods examined in the HCLIM2.5-MPI simulation, areas that have exhibited increasing trends in the present climate, were projected to experience an increase in ROS frequency in October, November and May, but a decrease is projected along the western and southern coast of Spitsbergen. In contrast, HCLIM-MPI projected an increase in ROS in January and February over these same areas. The projected decreases in ROS frequency across western areas close to the coast could be attributed to a later onset and earlier disappearance of the snow season and thus a shorter period for rain to fall on snow. This scenario is also supported by the results published in a recent report by Landgren et al. (2025) that show that considerable decreases in the fraction of spring and autumn precipitation falling as snow are projected in the future scenario along the west and southern areas compared to the present day. While in the present climate there are almost no ROS events anywhere across the archipelago during March and April (Vickers et al., 2024), the HCLIM-MPI climate projections indicate that, by 2050-2070 April could experience considerable increases in mean number of ROS events. In terms of potential for ground ice formation, April is at present the month with typically greatest snow depths, and ROS events occurring during April may not contribute significantly to formation of ground ice as thicker snowpacks will typically absorb rain before it reaches the snow-ground interface (Peeters et al., 2019). Moreover, if the onset of spring snowmelt also occurs earlier in the future, then many of the projected April ROS events may correspond to rain falling on an already isothermal snowpack, which would unlikely result in ground ice but rather increased surface runoff, with knock-on implications for fjord and marine ecosystems. However, since there were projected to be similar increases in ROS events across large parts of the archipelago during late autumn/early winter, the November ROS events may result in early formation of ground ice, thereby creating the potentially difficult foraging conditions for herbivores such as reindeer that persists throughout the winter. On the other hand, a later onset of snow and thinner early winter snowpack, coupled with more frequent and intense ROS could lead to complete ablation of the snow cover, providing greater access to forage. The future situation of ground ice formation due to ROS and its associated impacts on ecosystems is therefore challenging to foresee based on atmospheric parameters alone.

We have focused on interpreting the potential impacts of the changes in ROS characteristics projected by HCLIM-MPI due to its higher resolution offering better representation of the microphysical processes responsible for precipitation and extremes. However, the uncertainty in these projected changes arising from the use of just one climate model must be considered when drawing conclusions about future changes to ROS. While km-scale climate projections can reduce uncertainty by improving the model representation of key physical processes, their computational cost prohibits the development of a model ensemble to address uncertainty in downscaled projections. To partially address this, we analysed two additional HCLIM projections at the coarser resolution of 11km that downscaled two other CMIP6 climate projections for the SSP3-7.0 scenario which is a lower emission scenario than the SSP5-8.5 used in the km-scale simulations presented in this study (see Supplementary material). These two coarser resolution HCLIM projections downscaled the CNRM-ESM2-1 and NorESM2-MM simulations that were chosen using a storylines approach to address model uncertainty in the CMIP6 ensemble (Levine *et al.* 2024).

Our analysis of the coarser resolution HCLIM11-CNRM (Fig.S2) and HCLIM11-NorESM (Fig.S5) projections indicate that there are areas with a large degree of uncertainty associated with the magnitude and polarity of the projected changes. This uncertainty is demonstrated clearly by the HCLIM11-CNRM projections in Fig.S2 which projected decreases in all ROS characteristics over most parts of Spitsbergen, and especially considerable reductions in ROS intensity and total precipitation along the east coast of Spitsbergen for both future periods. These changes are opposite to those of the HCLIM-MPI and HCLIM11-NorESM datasets which projected increases over most of the archipelago, but with different magnitudes of change. This highlights the value in analysing projections with different physical storylines of future warming and addresses the range of uncertainty in the future ROS climatology in Svalbard and moreover demonstrates the potential scope of change which the ROS characteristics could fall within. These areas with greatest uncertainty, and hence least confidence are shown in grey in Figure S8 and correspond to the areas where there was a lack of agreement in the sign of change between the three models. It may be recognised that these are typically areas where ROS occur very infrequently in the present climate (Fig. 2), including the high elevation areas in the northwest of Spitsbergen and Nordaustlandet, and inland and eastern parts of Spitsbergen. There were nevertheless areas where we can have greater confidence in the projected changes, as shown by the areas where there was agreement across the three GCMs in parts of northern Spitsbergen and the northern coastal areas of Nordaustlandet for all four ROS characteristics. Moreover, it can be seen that in the 2050-2070 period, an increase in ROS frequency over northern and eastern parts of Spitsbergen and Nordaustlandet carries more certainty, as shown by the agreement across all GCMs.

Recent studies have analysed circulation-specific precipitation patterns in the present and future climate in Svalbard (Dobler et al., 2020) and find only small changes in the projected frequency of different weather types in the future climate, but nevertheless relatively large increases in precipitation. The authors found that the greatest changes in precipitation are projected to take place in the north and parts of the northeast of the archipelago, which was linked to higher precipitable water in the atmosphere as a result of reduced sea ice extent and greater evaporation. These areas also correspond to where our analysis found agreement in all three GCMs, with respect to increases in all ROS characteristics in the 2050-2070 period. On the other hand, all three GCMs analysed in this study projected an overall reduction in ROS duration, total precipitation and intensity in western parts of Spitsbergen which is unlikely explained by the same processes, or a reduced snow season length and could be

linked to changes in circulation patterns. While addressing the specific drivers of the observed patterns of change in ROS characteristics across the three climate models is outside the scope of the present study, further work could target determining the relative contributions of changes in sea ice concentration vs. remote (advected) moisture sources as drivers of the projected spatial variability of changes in ROS characteristics for a range of climate projections.

Lastly, the high elevation parts of the archipelago that exhibit greatest relative change in ROS characteristics in the HCLIM-MPI dataset have earlier been shown to be the same areas with greatest projected decrease in average winter snow depths (Isaksen et al., 2017), primarily the glaciated areas north and northeast of Svalbard. Sobota et al. (2020) studied the impact of ROS events over small glaciers in northwestern Spitsbergen and found an increase over the period 1976-2018 resulting in more ice layers, noting that ROS intensity in particular controlled ice layer thickness. Other studies of ROS impacts over glaciers have used snow model simulations or in-situ observations and shown that ROS events may increase the wintertime glacier mass balance as percolated water refreezes in the snowpack (e.g., van Pelt et al., 2016; Łupikasza et al., 2019), while a recent review of winter warm spells and heat waves highlights potential alteration of the glacier thermal regime when snow falls at relatively warm temperatures (Spolaor et al., 2025). Indeed, a recent study combined a set of ice core observations together with modelling of glacier stratigraphy and identified a transition in the thermal regime and stratigraphy of Austfonna in the northeast of Svalbard since 2013, from cold to temperate firn above the ice-firn interface (Innanen et al., 2025). In areas with large snow accumulation, multiple ROS events throughout the winter contribute to ice layers forming within the snowpack, which influences how rainwater from subsequent ROS events percolates through the snowpack, with knock-on impacts for runoff generation. Overall, the snowpack characteristics may be impacted by the significant increases in ROS that are projected to take place over Svalbard's glaciated areas within the next 50 years, although there nevertheless remains a lot of uncertainty in the polarity of change in some areas. While these areas may not be significant for land-based herbivores, increased melt and/or runoff from glaciers could have indirect ecological impacts by contributing with freshwater input to coastal or fjord environments, with subsequent impacts on fjord biogeochemistry and ecosystems (e.g, Vonnahme et al., 2023).

**4.3 Limitations of the study**

This study has utilized only one regional climate model to downscale one global climate model covering the high emissions SSP5-8.5 scenario. While this allows for an improved simulation of precipitation over smaller spatial scales which is crucial for mountainous and glaciated areas such as Svalbard, the use of only one RCM is a major limitation in capturing model parameterisation uncertainty and internal climate variability. Moreover, there remains a large degree of uncertainty in the magnitude and patterns of precipitation changes simulated by GCMs at local scales, limiting their use in risk assessments. These uncertainties have been in part demonstrated by our analysis of two additional 11km-scale climate projections, which showed that some areas of the archipelago that currently experience relatively few ROS events in the winter, carry greatest uncertainty, as exhibited by the opposite signs of change projected by the HCLIM11-CNRM and HCLIM11-NorESM projections (Fig. S2, S5). On the other hand, the areas which carried greater certainty in the projected sign of change were not constant across the two future periods analysed. The lack of agreement across the models across these areas also calls for a

more comprehensive analysis of the drivers of change, examining the relative importance of changes in snow season, and remote vs. local moisture sources. Future work should exploit an ensemble of models to produce a more robust projection of changes in ROS characteristics to identify similarities or discrepancies in the patterns of change across different models. It should also be emphasized that while the SSP5-8.5 scenario, represented by the km-scale HCLIM projections is a more extreme scenario based on current energy trends and socio-economic circumstances, it is nonetheless a plausible scenario and provides an upper bound on the range of possibilities and highlights the full range of risks for policymakers, so that they can understand what happens if mitigation measures fail.

Our approach utilizes daily values of meteorological (mean air temperature, precipitation) and surface (snow depth water equivalent) variables to detect days with rain-on-snow. These variables are commonly available across different reanalysis and climate model datasets which therefore makes the approach reproducible for other regions of the world where the impacts of ROS are consequential. However, a weakness in this approach is that the rain-snow transition is not properly represented. RCMs at km-scales offer a physics-based approach as the precipitation processes are represented exclusively by the microphysical scheme which separates precipitation into liquid and solid forms based on physics. This contrasts with coarser RCMs which must use both a microphysics scheme and a cumulus scheme, the latter of which cannot separate precipitation into liquid and solid form explicitly, thus necessitating the use of temperature-based approach. We have attempted to address this aspect by comparing the climatologies of each ROS characteristic for the 2000-2020 period using the HCLIM-ERA5 dataset, using temperature thresholds ranging from -0.5°C to 0.5°C, and when using the *prrain* variable, which gives 3-hourly estimates of the precipitation falling as rain (Figure A2). While there was better agreement between the *prrain*-based detection of ROS, and the temperature-thresholded approach using T=-0.5°C for the ROS frequency and duration for the archipelago as a whole, this temperature threshold produced much greater estimates of ROS total precipitation and intensity, especially in the western and southern parts of the archipelago. This is most likely an effect of taking a daily precipitation sum on days where the daily mean temperature exceeded this threshold. Whereas the *prrain*-based detection gives the total sum of rain on days with snow cover, the temperature threshold-based approach would most likely overestimate the total precipitation falling as rain, especially if there are wide variations in the daily temperatures. However, when examining the future changes of ROS in terms of the spatial variations, calculated using *prrain* (Fig. A3 to A5) the overall conclusions remain unchanged. Moreover, it should be highlighted that the temperature-based approach for partitioning rain and snow is nevertheless valid and necessary in cases where datasets have much coarser spatial resolution (eg >12 km) or temporal resolution and remains the only approach using ground observations where precipitation phase data are unavailable.

Hydrological modelling should also be considered, given that our analysis suggests a shift towards a large increase in ROS frequency in April. While April is a present typically a month with greatest snow depths and sea ice concentration around Svalbard, in a future climate April may be important for the onset of spring snowmelt, thus a large increase in ROS could also increase the potential for flooding impacts through enhanced runoff due to the combination of rain and rain-amplified snowmelt. There is also an uncertainty in how runoff may be affected by the presence of multiple ice layers within a snowpack caused by increased ROS frequency in areas with large snow accumulations. These impacts could be addressed in greater

detail to follow up the initial results presented here. Moreover, the potential link between sea ice concentration and ROS events
should be more carefully assessed and quantified, given the impact of continued rapid climate warming in the polar regions on
sea ice cover.
**5 Conclusion**
This study has examined five specific characteristics of rain-on-snow events across Svalbard in the present climate using the
CARRA reanalysis dataset, and in the future climate using high resolution 2.5 km-scale simulations from the HCLIM regional
climate model. ROS frequency, duration, total precipitation, intensity and seasonality were quantified. We found significant
and positive trends in all characteristics, but the significant trends were confined mainly to areas around the low-lying parts of
Nordaustlandet and areas in the east of the archipelago, while southern and western areas that typically exhibit greatest values
in all characteristics, were not found to exhibit significant trends in duration, total precipitation or intensity since 1991. Using
the same approach to quantify ROS, the HCLIM dataset driven by ERA5 input showed good agreement in the geographic
variability of all characteristics, though there were small differences in the absolute values. The HCLIM-MPI simulations for
2030 to 2070  project the greatest changes relative to present day conditions for all ROS characteristics in the mountainous
and glaciated areas to the north and northeast of the archipelago, while low lying western coastal areas are projected to decrease
in all characteristics.
We have attempted to address the uncertainty associated with using only one high resolution RCM and one GCM by analysing
two other HCLIM projections at the coarser resolution of 11km for the SSP3-7.0 scenario. Comparison of the km-scale SSP5-
8.5 HCLIM-MPI projections at 2.5km with these two coarser, SSP3-7.0. HCLIM projections show that there are large areas
of the archipelago where the change in ROS is uncertain, especially with regard to the magnitude and sign of change in the
ROS characteristics and demonstrated by the lack of agreement across the three GCMs analysed. However, these findings also
highlight the value in analysing projections from three GCMs that present different storylines of warming, specifically that the
range of uncertainty in future ROS climatology is addressed, as well as being able to identify areas where the changes are more
certain.
Changes in the length of the snow covered season when ROS can occur may affect the changes in frequency of ROS, but more
precipitable water in the atmosphere has been shown in similar studies to be a likely driver of increases in precipitation in
particular over northern and northeastern areas, where declining sea ice leads to greater surface evaporation . Lastly, our
analysis indicates that ROS have been increasing most in October, November, February and May, but in contrasting areas of
the archipelago. While ROS have increased most in the eastern and northeastern parts of Svalbard in October and November,
areas that are typically sensitive to sea ice concentration, it is primarily the western and southern parts of Spitsbergen that have
experienced significant increases in ROS frequency in February. The projections of Svalbard's future climate shows that there
could be an increase in ROS events across all months under a high emissions scenario, but substantial increases are also
projected to occur in January and April for the 2050-2070 period, months which currently experience very few, if any, ROS

events. This shift in ROS timing may have a cascade of impacts for both terrestrial and marine ecosystems that are dependent on snow and snowmelt.

## Data Availability

The CARRA dataset is publicly available and can be downloaded from the Copernicus Climate Data Store (CDS). The HCLIM 2.5 km regional climate simulations are available from https://thredds.met.no/thredds/catalog/pcch-arctic/catalog.html Until publicly available via ESGF, the 11 km HCLIM downscaled CNRM and NorESM datasets are available upon request to cordex@met.no.

## Author Contribution

HV and PM designed the study, OL prepared and made available the climate simulations, HV carried out the data analysis and prepared the manuscript with contributions from all co-authors.

## Competing Interests

The authors declare that they have no conflict of interest.

## Acknowledgements

This research was funded by Svalbard's Environmental Protection fund (grant 24/20) and the Framsenter research cooperation (incentive project CHORUS), and the Norwegian Research Council, project "PCCH-Arctic", under grant ID 320769. The climate simulations were carried out on the Norwegian Research Infrastructure Services (NRIS) high-performance computing facility "Betzy", operated by Sigma2, under project number NN9875K.

We acknowledge the support of PolarRES (grant no. 101003590), a project of the European Union's Horizon 2020 research and innovation programme. Storage and computing resources necessary to conduct the analysis were provided by Sigma2 – the national infrastructure for high-performance computing and data storage in Norway (project nos. NS8002K and NN8002K).

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

**Appendix/Supplementary information**

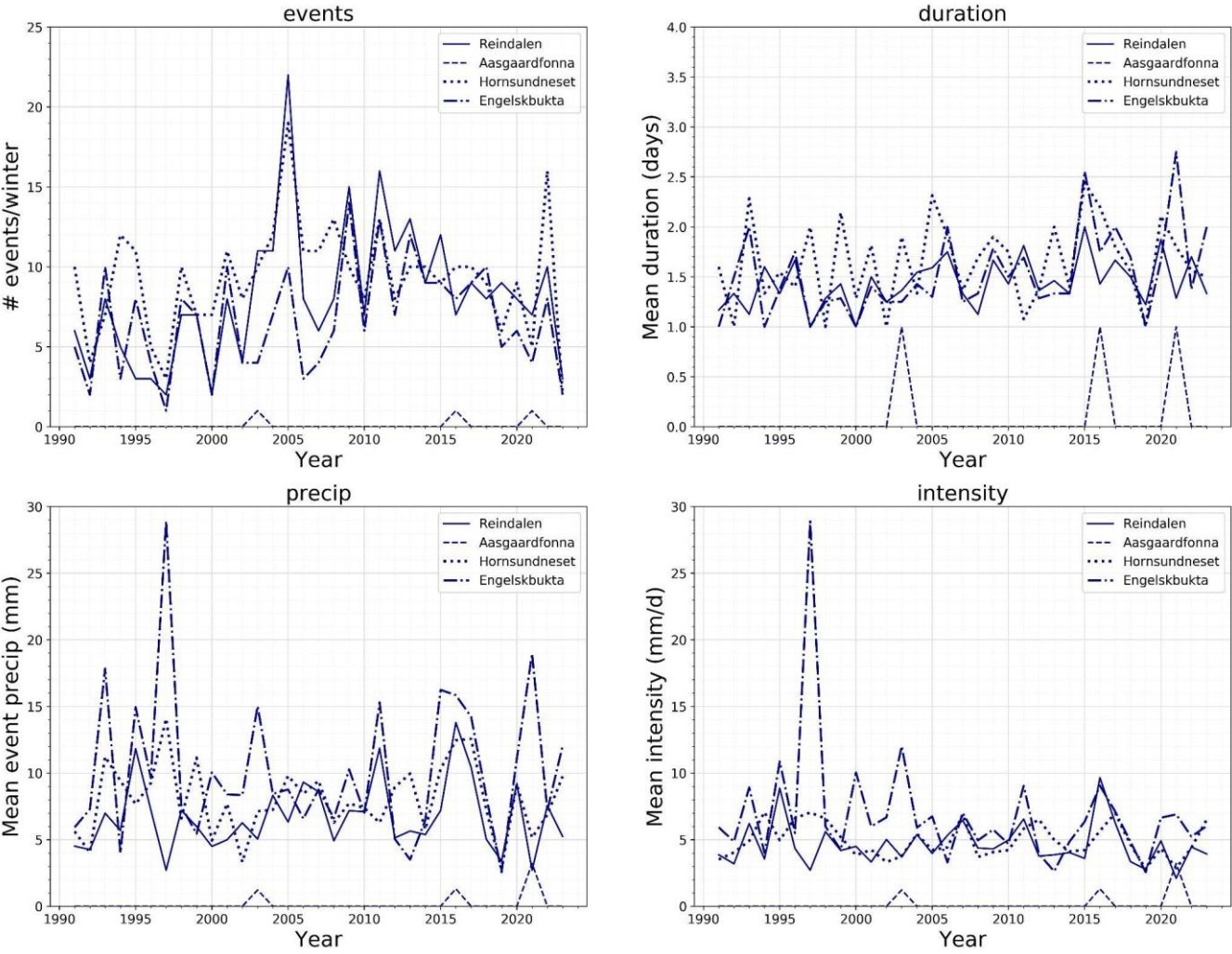

Figure A1. Time series of the ROS characteristics (number of events, mean duration, mean total precipitation and mean intensity) shown for four contrasting sites across Spitsbergen; Reindalen (77.98°N, 16.07°E), Åsgårdfonna (79.61°N, 16.61°E), Hornsundneset (76.88°N, 15.57°E) and Engelskbukta (78.84°N, 11.98°E). The coordinates correspond to one point within each area and are not exact coordinates for each site.

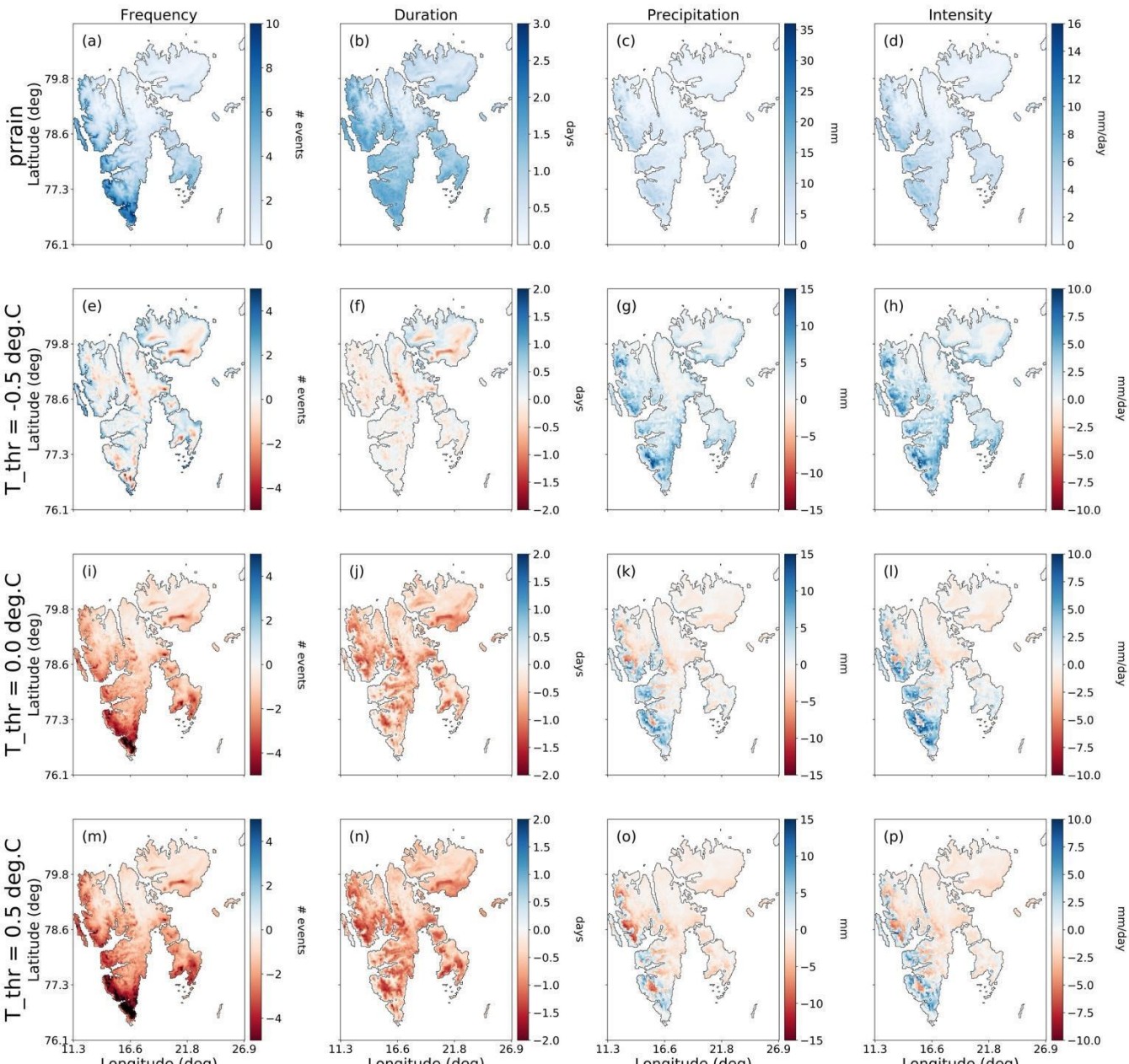

710

Figure A2. Mean ROS characteristics for the historical period (2000-2020) produced using the HCLIM-ERA5 dataset using the
accumulated rain variable (*prrain*) for detection, and the differences between the temperature-thresholded (t_thr) approach using
thresholds of -0.5°C, 0.0°C and 0.5°C The difference is given as ROS(t_thr) minus ROS(*prrain*) such that blue shades indicate areas
where the temperature-threshold approach produced higher values than *prrain*, and red areas indicate areas where *prrain* produced
higher values of the ROS variable compared to the temperature threshold detection.

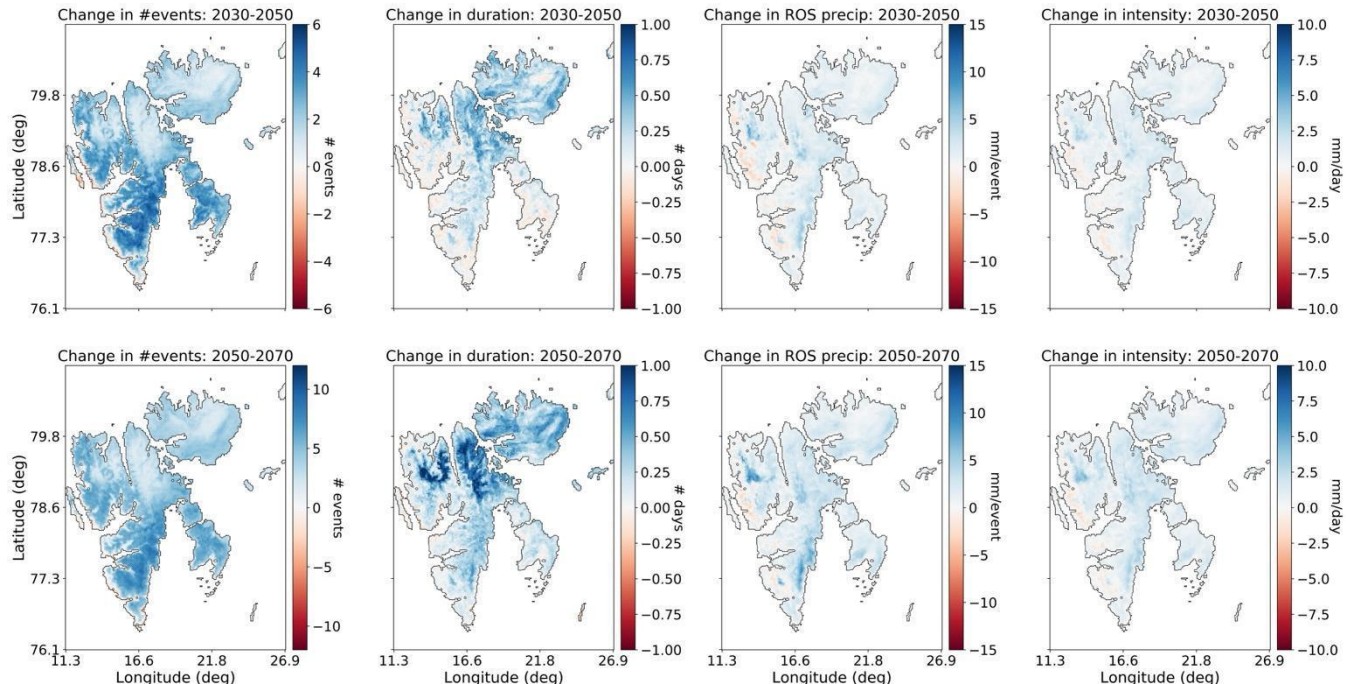

**Figure A3. Change in ROS frequency, duration, total precipitation, and mean intensity for 2030-2050 (upper row) and 2050-2070 (lower row) relative to the 2000-2020 averages, using the *prrain* variable to detect ROS.**

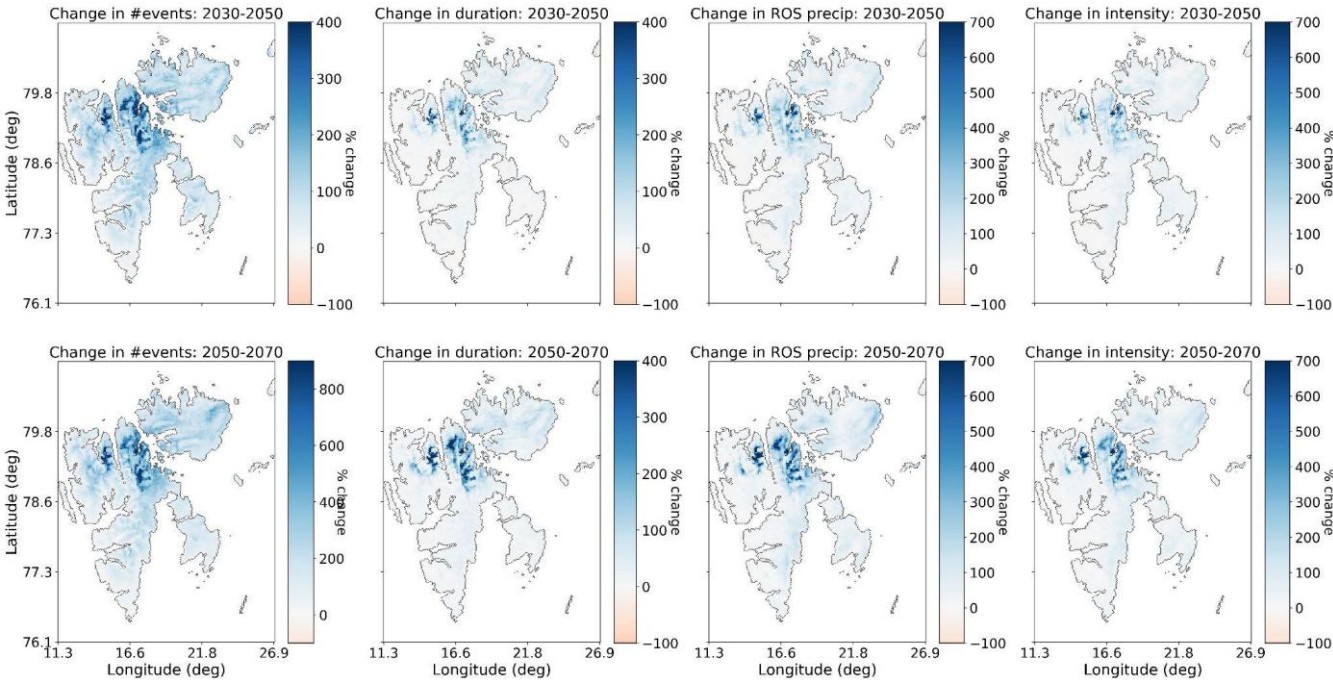

**Figure A4. As for Fig. A3 but with the changes expressed as a percentage of the 2000-2020 values.**

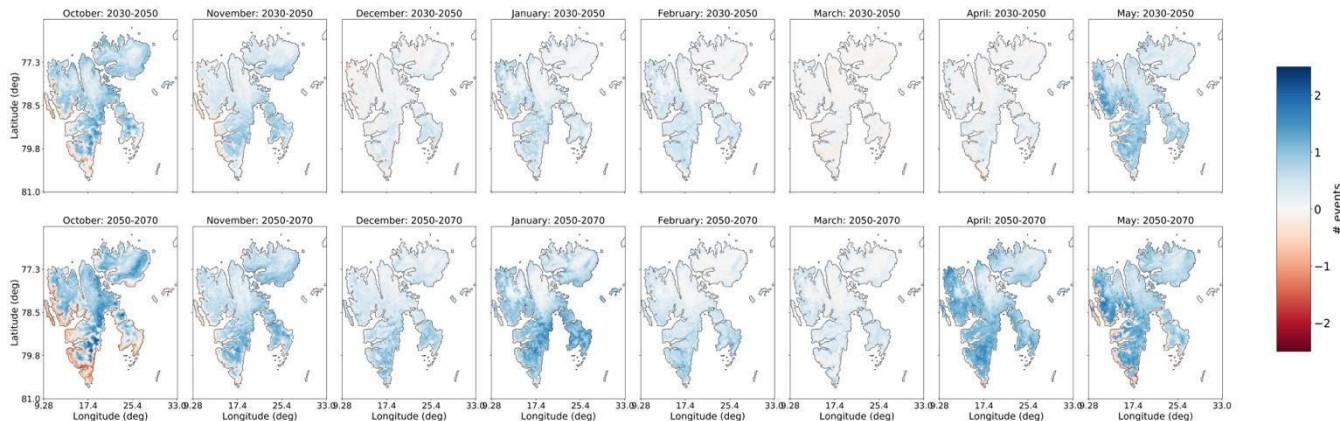

721

722   **Figure A5. Changes in ROS frequency by month from October to May for 2030-2050 (upper row) and 2050-2070 relative to the**
723   **reference period 2000-2020, when ROS were detected using the *prrain* variable.**