# Peer review of "Recent and projected changes in rain-on-snow event characteristics"

_EGUsphere, 2025_

## Author Response (AR1)

**Response to reviewers**

**Reviewer 1**
The manuscript describes a trend analysis study of rain-on-snow (ROS) events in Svalbard. The authors use reanalysis data (CARRA, ERA5) and one dynamically downscaled climate simulation (HCLIM-MPI) for this purpose and found, that ROS events have been increasing in frequency, duration, and intensity in several regions in the past, and are expected to further increase in the future. To motivate this study, the authors provide a brief discussion of the ecological and hydrological consequences of ROS events and propose directions for future research.

Major comments:

A convection permitting RCM (HCLIM) is used for downscaling the single GCM used in this study. The authors argue, that this would lead to superior results compared coarser-scale RCMs (e.g., l74, l363), but miss to demonstrate this claimed added value. One reference is given (Landgren et al., 2025), but this is only a conference abstract with no information content (no results in the abstract). I.e., a basic evaluation and demonstration of added value of the HCLIM is completely missing, which leads to several complications. One of them is, that if no significant added value compared to other RCMs (e.g., from CORDEX-ARC-22) can be demonstrated, the authors could use an ensemble of conventional climate models instead. This would resolve the major weakness of this study (see following comment). Another issue of missing model evaluation is that the authors have to speculate on the reasons for differences between CARRA and HCLIM ROS characteristics. E.g., l312: "The discrepancy in absolute values of the characteristics for the present climate (2000-2020) are likely attributable to the uncertainty in temperature threshold for partitioning rain and snow used in the different datasets". Couldn't it also be a simple temperature bias in HCLIM?
*A recently published report that provides a comprehensive evaluation of the HCLIM simulations (https://www.met.no/publikasjoner/met-report/_/attachment/inline/b1b9b27c-4086-452f-a8e7-b7bdbeb4053f:996133826c7a1b2d83d35602f425fedbad34f525/MET-report-01-2025.pdf) indicates that compared to CARRA, HCLIM tends to produce a slightly colder bias over land over Spitsbergen, but slightly warmer bias over eastern areas due to lack of snow cover over sea ice in ERA5 in the Barent sea region. This certainly could contribute to the differences we observed in the ROS characteristics between CARRA and HCLIM-ERA5, with underestimations in the characteristics by HCLIM over Spitsbergen but smaller differences over the northeastern areas (Nordaustlandet). Based on this we will amend our discussion to highlight this point in explaining the observed differences.*
My main concern about this manuscript is the use of only one GCM/RCM combination for the analysis of future trends. Particularly, the choice of the GCM can be expected to have large impact on the results. This weakness is clearly identified by the authors (l364), which is good, but at the same time, this is no justification, since other options would be available. E.g., the CORDEX-ARC-22 archive contains 79 simulations (https://esgf-node.ipsl.upmc.fr/search/cordex-ipsl/). The authors have to demonstrate how this single realization be regarded as representative, or at least show whether is a cool/warm and dry/moist realization of expected future climate. Or, preferably, use a comprehensive model ensemble, as it is state-of-the-art.

*There are 79 simulations when considering all variables and all scenarios together. When these are filtered for 2-metre temperature (tas), there are only two RCMs (CanRCM4 and CRCM5) available for the ERA5-driven historical period, and only one RCM (CanRCM4) available for future scenarios. Since this simulation is almost 10 times coarser in spatial resolution (0.25 degrees) and Svalbard is an area of very steep topography which drastically reduces the usefulness of data from coarser simulations, we did not want to consider it for this study. This need for finer scale resolution is already evident in our results which show that variations in ROS and its characteristics can vary a lot with topography and thus such variations would be missed in the 0.25 degree CORDEX simulations. However, we have now included an analysis of 0.11 degree Polar CORDEX simulations from 2 additional GCMs that have been downscaled by the same RCM which have not yet been published on ESGF in a separate Supplementary Materials section, so the inclusion of these simulations should improve the robustness of the study as well as address the range in uncertainty of future ROS climatologies. Moreover, these simulations use the SSP3-7.0 scenario, which also broadens the potential future ROS scenarios.*

The method of trend detection is not named nor described. Please give a clear explanation of the statistical method used. This is particularly important, since the analyzed period (1991 – 2023) is rather short for detecting significant trend in time series with large variability. I additionally suggest presenting some time series, in order to allow the reader to get an impression of the variability involved.

*The trend was calculated by performing a linear regression, using the slope of the linear model to give the annual trend and converted to a decadal trend. This has now been clarified in the text. In our revisions we have now included time series of the four ROS characteristics from four different sites across Spitsbergen to illustrate ROS characteristics representing contrasting inland and coastal climates as well as latitudinal variations. This is shown in figure A1.*

l197: the results show a maximum of the trends in November. This would imply, that significant trends could also be present in October, or even earlier in the year. Please expand your evaluation time-window (currently Nov-April) accordingly, in order not to miss significant results.

*The November to April period was originally chosen based on an earlier study where SAR wet snow detection had been used to detect ROS (Vickers et al., 2022) and was subsequently evaluated against CARRA (Vickers et al., 2024). This period was necessary because snow cover could be transient in October and May is a month when spring snowmelt begins at low elevations, so the SAR approach is less useful to detect ROS in these months. However, we have now extended the analysis period from October to May for all datasets used since the reanalysis dataset is not subject to the same limitations as SAR. All figures have been updated based on the new period.*

l258: The future trends feature a maximum in Nov., a minimum in March and another maximum in April. This seems to be counter-intuitive and would deserve some discussion, or ideally an explanation.

*We expect further warming to lead to higher probability of ROS at the tail ends of winter when the average air temperature may be warm enough that rain falls instead of snow. Moreover, warmer conditions in the future in November will likely lead to later onset of snow cover in low lying coastal areas (i.e. November may become the new October with a transient snow cover, therefore fewer ROS days will be detected if no snow cover is present). In a warmer climate there will be a shift to more frequent ROS at higher elevations*

*but potentially lower at the coast where snow cover may appear later than in the present climate. We have included in the discussion regarding April trends that warming could lead to higher probability of rain and/or potentially earlier spring snowmelt*

Minor and editorial comments:

The introduction features some repetitions (e.g., the fact that ROS impact ecosystems) and would gain from streamlining/shortening.
*We have reviewed the introduction and excluded repetitions where relevant*

**Reviewer 2**

This is a useful paper examining recent and future changes in rain on snow (ROS) events and their characteristics over Svalbard. ROS events in the Arctic have generated much attention in recent years due to their many impacts. This paper is a solid contribution. I have only a few comments and suggestions.

Specific

Abstract and elsewhere: It is widely viewed that SSPS-8.5 is an overly aggressive scenario. This needs to at last be discussed in the text.

*Yes. We agree that SSP5-8.5 is a more extreme scenario based on current energy trends and socio-economic circumstances. Nonetheless, it is a plausible scenario and provides an upper bound on the range of possibilities. As such it highlights the full range of risks for policymakers, so that they can understand what happens if mitigation measures fail. We have now also explored using simulations at a coarser resolution (11km) that are based on a more moderate emissions scenario (SSP3-70) and discussed the results from these simulations in the context of uncertainty in the future change in ROS characteristics.*

Line 11: It is "specifically" duration, intensity seasonal timing, yes? (not "such as")

*Yes, and we have changed the wording to "specifically"*

Line 16: Following from the above comment, by "all characteristics" do you mean frequency, duration, and intensity? Please be specific.

*Yes, we have emphasised specifically the characteristics referred to*

With respect to CARRA, why only go back to 1991? Was it not run for earlier years?

*No, the present version of CARRA is only available from 1991 (data from 1990 was also available but not for a full year). The forthcoming edition of CARRA2 will provide reanalyses dating back to 1985.*

Line 28: It is important to note that "Arctic Amplification" is very seasonal - largest in autumn and winter, and smallest in summer.

*We emphasise this point in the revised version*

Line 91: 121.8 mm is extremely dry. For the reader not familiar with Arctic precipitation, it would be useful to point this out.

*We have included this detail in the revised manuscript*

Section 3 and elsewhere: A pet peeve of mine: trends are positive or negative, not increasing or decreasing. In increasing trend implies that the trend is getting larger.

*We have changed instances of increasing/decreasing to positive/negative*

A question on Figure 3: Is part of the reason what there are no trends in April because over at parts of Svalbard, there is no snow on the ground? I think the answer is no, but a reader may be wondering. A clarifying sentence is warranted.

*There are trends in all months, but we have chosen to only show where the trend was at least marginally significant i.e we have masked out areas with confidence > 0.1. So the lack of trend values in Fig.3 doesn't mean no trends, it just means they were too weakly significant to be included.*

With respect to Figure 4, the paper could benefit from more discussion of the causes of precipitation changes. As discussed later in the paper, proximity to sea ice may play a role, but what about the idea that in warming climate the atmospheric carries more water vapor? Furthermore, there ae negative trends along the west coast, which argues for changes in circulation patterns. Also, what are the projected changes in sea ice? I assume that the ice margin by 2050-2070 has retreated well to the north.

*We have currently argued that the negative trends in ROS frequency close to the coast are more likely the result of shorter snow cover duration i.e. later onset, earlier snowmelt -> shorter snow cover = reduced time period when ROS can occur. This scenario is also supported by the results published in a recent report by Landgren et al. (2025) that show there will considerable decreases in the fraction of spring and autumn precipitation falling as snow in the future scenario along the west and southern areas compared to the present day (this was not included but will be in the revised manuscript). In our revised manuscript we have broadened our discussion to include the possibility that the areas with observed increases in ROS can be due to increased precipitable water in the atmosphere as a result of declining sea ice surrounding those regions. However, we agree that the reductions in ROS precipitation and intensity along the west coast are unlikely due to this explanation, but it was outside the scope of this work to determine the changes in drivers of ROS.*

---

## Author Response (AR2)

Editors report 7 November 2025

Dear authors

Thank you very much for the thorough revision. I have asked one of the referees, and I agree with their assessment that the manuscript has improved substantially. The referee has only one remaining concern. Because the newly included additional climate model runs show results that are opposite to those produced by the model used previously, the referee suggests making this uncertainty even more prominent throughout the manuscript (in the abstract and the results).
Alternatively, please mention this new outcome in the abstract and also discuss it in the "Limitations of the study" section.

In response to the recommendations made by the reviewer and Editor, we have now included additional text in the abstract and sect. 4.3 "Limitations of the study" to emphasise the uncertainty in the projected changes of ROS across Svalbard as demonstrated by the results of our analyses of two additional 11km-scale projections. We hope these changes are satisfactory.